# Can We Manage Behavioral Problems through the Development of Children’s Social-Emotional Regulated Behavior? Longitudinal Study of a Preschool Program

**DOI:** 10.3390/ijerph18168447

**Published:** 2021-08-10

**Authors:** Ana Justicia-Arráez, María Carmen Pichardo, Miriam Romero-López, Guadalupe Alba

**Affiliations:** Department of Developmental and Educational Psychology, University of Granada, 18071 Granada, Spain; pichardo@ugr.es (M.C.P.); miriam@ugr.es (M.R.-L.); guadalupe@ugr.es (G.A.)

**Keywords:** externalizing problems, socio-emotional competence, social skills, regulated behavior, emotional regulation, childhood, preschool, universal intervention, prevention

## Abstract

Behavioral problems are early indicators of antisocial behavior and should be targeted from a preventive perspective from early childhood. The purpose of the study was to analyze the effectiveness of the AC1 preschool program that develops social-emotional skills that facilitate the adjustment and regulation of the person. A total of 102 children aged 3–4 years old participated in the research, 52 belonging to the experimental group and 49 to the control group. Program-trained skills (ROAC-3), social skills (PKBS-2), and externalizing problems (CBCL C-TRF) were assessed in the pre- and post-intervention phase. Data analysis was carried out using a generalized linear mixed model analysis (GLMM). The results show that the children in the experimental group scored higher on the variables trained by the program and on social skills than those in the control group. They also obtained lower scores in the observed externalizing problems. The effect of the program was high in the emotion identification and expression, communication skills, prosocial behaviors (sharing and helping), problem-solving, and social interaction. Social-emotional learning in early childhood is essential for the prevention of behavioral problems to facilitate the development of adjusted and regulated behavior. Thus, preschool programs could play a key role.

## 1. Introduction

Childhood behavior problems are considered early indicators of antisocial behavior, especially aggressive and destructive behavior [1,2]. They have been characterized as persistent and repetitive patterns of behavior in which the basic rights of others, norms, or age-appropriate social rules are not respected [3]. Specifically, externalizing problems are uncontrolled behaviors that the person directs outward or toward others such as attention problems, aggressive behaviors, or hyperactivity.

These types of problems have aroused great interest within the scientific field because they generate devastating consequences for the individual and society [4] and also in the educational field because they affect the learning process, academic performance, classroom climate and social relations [2,5,6], self-regulation [7,8], and they have even been related to subsequent involvement in other phenomena such as bullying [9].

The prevalence rate of externalizing problems varies from one country to another depending on the type of problem and also on the measuring instrument. In Spain, there are studies that place the rate of the child and adolescence conduct problem between 10 and 15% [10]. Moreover, the latest National Health Survey indicates that the prevalence of conduct disorders in the population aged 0–14 years was 1.8%, which has remained stable since 2006 [11]. 

Several studies have indicated that behavioral problems in early childhood have a certain stability over time [12], which is usually higher after four years of age and tends to increase [13,14]. In addition, the persistence over time of behavioral problems can lead to other more serious pathologies such as aggression, delinquency and antisocial behavior [15], inattention, hyperactivity, anxiety or inability to help others [16] as well as other types of mental health, educational, social, occupational, or physical problems [17]. 

Therefore, effective early intervention is necessary so that behavioral problems do not worsen over time [18]. In this sense, the preschool years are a key time for the prevention of behavioral problems. From this preventive approach, we should highlight that a large part of the intervention programs are focused on the development of adaptive behavior, specifically social and emotional competence [19]. 

Social competence has been defined as the ability to manage thoughts, feelings, and actions according to personal goals and the demands of the situation and culture [20] and the dimensions in preschool including problem-solving, peer relations, compliance with rules, and frustration tolerance [21]. On the other hand, emotional competence is defined as the ability to adequately manage the emotions that arise in social transactions [22]. In preschoolers, this competence involves the acquisition of emotion knowledge (expressive/receptive recognition), emotion expression (predominantly positive expression), and emotion regulation (adaptive self-regulatory strategies) [21,23].

Social and emotional learning involves the development of five components that include a variety of thoughts, behaviors, and attitudes [24,25]: (1) self-awareness, to understand one’s emotions, thoughts and actions; (2) self-management, that is, self-regulatory capacity linked to emotions, impulse control, delay of gratification, stress management and goal attainment; (3) social awareness, related to empathy, social perspective-taking, or understanding of social norms; (4) relationship skills necessary for the maintenance of healthy relationships in accordance with social norms; and (5) responsible decision-making at the personal and social level adjusting to the context. 

As can be seen, social and emotional competence requires a high level of management and regulation by the individual. Self-regulation strategies play an important role in social and emotional performance, since the person must plan, supervise, and adjust their behavior in changing social circumstances and emotional states. Accordingly, some authors have pointed out that the construct of self-regulation encompasses competencies that include the control of emotions or positive relationships with others [26]. In fact, the positive association between self-regulation and social skills [26,27] and the negative relationship with externalizing problems has been demonstrated [8,28]. Similarly, early self-regulation skills predict positive school adjustment [29].

Research on the evidence for the effectiveness of programs that promote socio emotional learning (SEL) has been growing in recent years and different studies have highlighted its benefits. A recent review of SEL programs in early childhood [30] points to the effectiveness of some of these interventions in improving social-emotional competence [31,32], cognitive regulation [33,34], interpersonal problem solving [35], and academic performance [33] as well as in reducing conduct problems and aggressive behaviors [34].

As far as we know, there are only a few evidence-based programs that develop these types of competencies in the Spanish preschool population. It is a field that is in full development, but there are still not many studies with data on the effectiveness of the programs. This is evident in the publication of various meta-analyses [36,37], which show that most of the programs that have evaluated their effectiveness are American or Canadian [38]. 

Therefore, it is essential to carry out studies with a Spanish population to analyze the efficacy of intervention programs focused on preventing behavioral problems. Research in this field is essential, especially if we take into account that in Spain, the schooling rate at three years old is 96.1% [39]. In this sense, and coinciding with Greenberg et al. [25] on the need to consider SEL as a public health approach to education in order to improve the general population’s wellbeing, schools become ideal spaces for developing this type of competencies and skills, since most children spend a significant time each day at school. 

In order to contribute to this field, the purpose of this study was to provide evidence of the efficacy of the Aprender a Convivir program, a universal preschool prevention program focused on improving various skills related to social and emotional competence and reducing conduct problems. Preliminary studies in which there was no control group have already shown the positive effect of the program on social competence and behavioral problems in 3-year-old children [40]. On the other hand, there are studies that have shown the effect of this program on the improvement in social skills and the reduction in internalizing and externalizing problems in 4-year-old children [41]. In addition, its impact has been analyzed on social competence after three years of intervention [42]. 

Specifically, this study focuses on the impact of the program in 3–4-year-old children and the following objectives are proposed: To present and analyze the effects of the aspects that the program trains (rules, emotion identification and expression, emotion regulation, communication skills, sharing and helping, sharing personal belongings, and problem solving). It is expected that the students who participate in the intervention will improve these skills significantly compared to the control group.To analyze the impact of the program on social skills and externalizing problems. In this case, the objective is to check if there is a transference and the program improves other types of social skills that require greater self-regulation when interacting with the behavior of others such as social cooperation, social interaction and social independence, and it reduces externalizing problems. In this sense, it is expected that children in the experimental group will improve in these variables compared to those in the control group.

## 2. Materials and Methods

### 2.1. Participants

A total of 101 children aged 3–4 years (M = 42.55 months, SD = 3.42) participated in the study, 44.6% boys and 55.4% girls. The pupils were enrolled in the first year of the second cycle of preschool according to the Spanish educational system and they came from two schools with similar characteristics located in Granada, Spain. Both schools are semi-private schools (private school that received public funding), located in the same metropolitan area (they are close to each other), have the same enrollment capacity, and a similar educational approach. 

The schools were randomly assigned to the experimental condition. The experimental group included 52 children, 22 boys, and 30 girls (M_age_ = 42.64 months, SD_age_ = 3.57) whereas the control group was formed by 49 children, 23 boys and 26 girls (M_age_ = 42.46 months, SD_age_ = 3.31). All participants were Caucasian and middle socioeconomic level. None of the participants had relevant special educational needs.

### 2.2. Instruments

The Observation Scale of the Aprender a Convivir program for 3-year-old children (ROAC-3) [43] was used to assess the variables trained by the program. It is a Likert questionnaire with response values ranging from 0 (never) to 4 (frequently). It is structured by 26 items grouped in seven factors: (1) rules (six items), associated with following the rules established in class (e.g., “Tidies up materials after activities”); (2) emotion identification and expression (emotion IE) (four items) (e.g., “Differentiates between happiness and sadness”); (3) emotion regulation (two items) (e.g., “Controls his or her feelings”); (4) communication skills (four items), associated with basic interpersonal communication skills such as listening, asking for the word or giving thanks (e.g., “Keeps quiet when others are talking”); (5) sharing and helping (four items) (e.g., “Helps peers when needed”); (6) sharing personal belongings (two items), a factor that assesses sharing behavior (e.g., “Protests when she/he has to share her/his materials”); and (7) solving problems (four items), which assesses how the child acts when a conflict arises (e.g., “Talks to solve problems”). The original instrument shows adequate values and fit indexes, and the internal consistency is adequate in the Spanish preschool population [43]. 

Social skills were assessed using the Spanish version [44] of the Preschool and Kindergarten Behavior Scale for Teachers and Caregivers (PKBS-2) [45]. It is a Likert scale with response values ranging from 0 (never) to 3 (often) composed of 34 items grouped into three factors: (1) Social cooperation (12 items) assesses important skills needed to follow adult instructions, to cooperate and compromise with peers, and to show self-control at the same time (e.g., “Shows self-control” or “Responds appropriately when corrected”); (2) social interaction (11 items) measures behaviors necessary to gain and maintain acceptance and friendship from others (e.g., “Try to understand another child’s behavior”); and (3) social independence (11 items) assesses behaviors such as accepting temporal separation from adults or having confidence, which encourages the child to acquire a social independence that promotes personal autonomy (e.g., “Adapts well to different environments” or “Is confident in social situations”). The Spanish version of the PKBS-2 shows adequate values and fit indexes and the internal consistency is adequate [44,46]. 

Externalizing problems were measured with the Child Behavior Checklist for ages 1_1/2_–5 Caregiver–Teacher Report Form (CBCL 1_1/2_–5, C-TRF) [47]. This instrument measures two dimensions of behavioral problems (internalized-externalized). Only the externalizing problems scale was used for the present research. It is a Likert scale with response values ranging from 0 (not true) to 2 (often true). It is composed of 34 items that assess the attention problems related to dispersion of thought and difficulty to focus on something (e.g., “Quickly shifts from one activity to another”) and aggressive behavior involving disruptive behaviors directed at people or objects that may cause physical, psychological, or social harm (e.g., “Temper tantrums or hot temper”). The internal consistency indexes obtained with the study sample were adequate (α = 0.91).

### 2.3. Aprender a Convivir Program

This program is an educational intervention published in three volumes corresponding to the second cycle of preschool (3–6 years), in accordance with the Spanish educational system. In this research, only the Aprender a Convivir 1 program (AC1 program) was used, aimed at children aged 3–4 years [48]. The aim of the program is to develop student’s social skills in order to prevent the appearance of behavioral problems. It consists of 12 sessions that are applied twice a week with a duration of 45 min and organized in four blocks of content: (1) Rules and compliance with them, where we work on rules of coexistence and interpersonal relationships as well as values; (2) Feelings and emotions, which deal with the identification, expression, and regulation of basic emotions; (3) Communication skills, focused on developing skills such as listening, asking for the floor, asking for things please or saying thank you; and (4) Help and cooperation, which promotes a series of prosocial behaviors such as sharing, helping, or cooperating. In a cross-cutting way, in each of the blocks, the regulation and resolution of problems is worked on. In addition, work is carried out in a targeted manner with the family, proposing activities to be carried out at home every 15 days.

The program presents an active methodology focused on the use of puppets, which are responsible for transmitting the main objectives in each session. Dialogue and the expression of ideas are encouraged and various resources such as stories, songs, games, and activities are used. Sessions generally have the same structure: (1) introduction with puppets; (2) activity (and an alternative activity); and (3) home activity (and an alternative home activity).

### 2.4. Procedure

The study followed the protocols established by the schools and with the current Spanish data protection law to obtain parental permission and informed consent. The research was also conducted according to the ethics regulations of the University of Granada.

To initiate the research, one of the school areas of the city of Granada (Spain) was randomly selected and all the preschool centers were contacted. Among the schools that confirmed their interest, a random selection of two schools that finally took part in the research was made. Subsequently, the schools were randomly assigned to the experimental and control condition, and the permissions of the schools and the consent of the families were obtained. Prior to the start of the research, an information meeting was held for the teachers and families of the classes involved. Those who could not attend received an information letter with all the details of the study.

A series of phases were followed in the study (Figure 1). First, an external evaluator was trained to observe and assess the study participants (pre-intervention phase). During this phase, students were observed at different times at school and the person in charge was provided with a register in which the frequency of the behaviors under analysis was registered. Once the observation was completed, the external observer filled the scales used in the study. In the meantime, a teacher from outside the research was trained to carry out the AC1 program. Two members of the research team with academic training in early childhood education and psychopedagogy carried out the training. Two 5-h training sessions were held. In the first, the AC1 program was presented: objectives, contents, methodology, activities, resources, and evaluation system. The second session, of an applied nature, was aimed at the elaboration of the puppets and the presentation by the teacher of two sessions of the program. A blind procedure was used as the observer and the teacher were unaware of the objectives of the study. 

This was followed by the intervention phase carried out by a teacher, which lasted 12 weeks. In the experimental group, the AC1 program was implemented, and in the control group, there was no intervention of any kind related to this topic. The intervention sessions were held two days a week for 45 min. 

Four weeks after the end of the program (latency period), the post-intervention phase began. The external evaluator observed and assessed the behavior as in the pre-intervention phase. 

### 2.5. Analysis

Taking into account the objectives of the study, an individual randomized trial experimental study was designed with two groups (experimental and control) and two evaluation phases (pre-intervention and post-intervention). 

Initially, descriptive analyses are carried out in which the means and standard deviations of the variables considered are presented. Given the distribution of the data, and that this was a repeated measures study, the generalized linear mixed model (GLMM) analysis with logit link function with Poisson distribution with a random intercept for each subject was used to compare the two groups on the primary outcomes at two time points. The independent variables in the model included a binary variable for group assignment (intervention vs. control), a binary variable for time (pre-intervention vs. post-intervention), and their interaction term. Comparisons were made between the control and intervention groups in the two phases considered (pre-intervention and post-intervention) using a t test with sequential Bonferroni adjustment. Finally, the effect size of the differences between the experimental and control groups is included. Considering that there was no normal distribution of the data for the different variables, a non-parametric mean difference test (Mann–Whitney U) was performed to obtain the effect size of the intervention, using the values of this non-parametric test to obtain Cohen’s d [49]. Cohen established large (d ≥ 0.80), medium (0.50 ≤ d ≤ 0.79), and small (0.20 ≤ d ≤ 0.49) effects [50].

The Statistics 24 for Mac version of the Statistical Package for the Social Sciences (SPSS) (IBM, Armonk, NY, USA) was used for the different analyses carried out in the study.

## 3. Results

Table 1 shows the descriptive measures for the outcomes used in the AC1 program, differentiated by groups (experimental and control) and time (pre-intervention and post-intervention). As is shown, the participants in the experimental group started with higher mean scores than the participants in the control group in all the trained skills variables except emotion regulation and sharing personal belongings. Similarly, both groups increased their mean scores in the post-intervention phase. 

Table 2 shows the descriptive measures of the social skills and externalizing problems of the groups differentiated by intervention time. As the means of the social skills and externalizing problems considered show, the participants in the experimental and control groups obtained similar scores in the pre-intervention time. Taking into account the mean scores obtained by both groups in the post-intervention phase, the participants of the experimental group increased their scores in social cooperation, social interaction, and social independence, and reduced their scores in externalizing problems. Participants in the control group, on the other hand, obtained very similar scores in both phases. 

Initially, comparisons between groups (experimental, control), time (pre-intervention, post-intervention), and interaction group*time were run. The analyses of the preliminary fixed-effect comparisons are shown in Table 3. Significant differences were observed when comparing by group in rules, emotion IE, communication skills, sharing and helping, sharing personal belongings and solving problems. On the other hand, differences were found between the pre-intervention and post-intervention phases in all program outcomes. Related to the group*time interaction, significant effects were observed in emotion regulation, communication skills, sharing personal belongings and solving problems. 

Later, predictive analysis, carried out by taking into account the random effects of each participant in each of the phases assessed, showed significant differences between the experimental and control groups in rules, emotion IE, communication skills, sharing and helping, and solving problems. The contrast test between pre-intervention and post-intervention times showed significant differences in the same variables. Finally, the interactions group*time were significant in emotion regulation, communication skills, sharing personal belongings, and solving problems.

The pairwise contrast, carried out by the model using the estimated means, taking into account the random effects as it is a repeated measures model (see Table 4), indicates the existence of significant differences between the experimental and control group, in which the experimental group obtained a higher score in rules, emotion IE, communication skills, sharing and helping, and solving problems. Similarly, there are differences between the two groups, with the control group scoring higher in sharing personal belongings. On the other hand, there were no differences in emotion regulation. Analyzing these contrasts by pairs in the comparison over time (pre-intervention-post-intervention), the results showed significant differences between both times in all variables. In all of them, the scores obtained in the post-intervention time were higher than in the pre-intervention. 

Table 5 shows the pairwise contrasts for the interaction group*time, taking into account the estimated means of the random effects model. The results showed that in both groups, there were significant differences between the pre-intervention and post-intervention phases in rules, emotion IE, communication skills, sharing and helping, and solving problems, with higher scores in the post-intervention phase in both groups. However, the differences between both phases were greater in the experimental group in all variables. Nevertheless, in the variables emotion regulation and sharing personal belongings, the differences between both times were only significant in the experimental group. In addition, in these variables, the scores of the participants in the experimental group were higher in the post-intervention phase.

Figure 2 shows the pairwise contrast for the group*time interaction taking into account the means estimated by the model on the program’s variables.

In order to analyze the effect of the intervention program, the effect sizes between groups were calculated using the non-parametric Mann–Whitney test. The results showed a low effect size for rules (d = 0.37) and emotion regulation (d = 0.07), medium effect size for sharing personal belongings (d = 0.59), and high effect size for emotion IE (d = 1.36), communication skills (d = 0.83), sharing and helping (d = 0.82), and solving problems (d = 1.42).

On the other hand, in order to contrast the effects of the program not only on the variables trained by the AC1 program, but also on the social skills and externalizing problems, a generalized linear mixed model was also carried out considering both groups and intervention times. The results of the analysis are shown in Table 6. The initial results of the fixed effects analysis showed significant differences between the experimental and control groups in the social skills factors: social interaction and social independence. There were no differences in social cooperation and no differences in externalizing problems. On the other hand, the results showed significant differences between pre-intervention and post-intervention times in all social skills variables (cooperation, interaction and independence) and also in Externalizing problems. Finally, regarding the group*time interaction, the results show significant interactions in all the variables analyzed.

Later on, the predictive analysis carried out, considering the random effects of each participant in each of the phases evaluated, showed significant differences between the experimental and control groups in social cooperation, social interaction, and social independence. Similarly, differences were also observed between the two groups in externalizing problems. In the case of social skills, the experimental group obtained higher scores in all variables. On the other hand, in externalizing problems, the experimental group obtained lower scores than the control group. The analysis of differences between the different evaluation moments (time) showed significant differences between pre-intervention and post-intervention in social cooperation, social interaction, and externalizing problems. In the case of social cooperation and social interaction, scores were higher in the post-intervention phase. In contrast, in externalizing problems, the score was higher in the pre-intervention phase. As for the group*time interaction, significant interaction effects were observed in all social skills variables and also in externalizing problems. 

The results of the pairwise contrast according to groups (experimental and control) and time (pre-intervention and post-intervention), carried out by the model, using the estimated means, taking into account the random effects, can be seen in Table 7. Differences between groups were observed in social interaction and social independence, with higher scores in both groups in the experimental group. On the other hand, there were no significant differences between the two groups in social cooperation and externalizing problems. Related to the differences between the pre-intervention and post-intervention, differences were observed in all the variables analyzed. In social cooperation, social interaction, and social independence, the scores in the post-intervention were higher than in the pre-intervention. In the case of externalizing problems, the highest scores were found in the pre-intervention phase. 

Pairwise contrasts on the interaction group*time, taking into account the estimated means of the random effects model (see Table 8) showed differences between the two-time phases in the experimental group in social cooperation, social interaction, social independence, and externalizing problems. Participants in the experimental group obtained significantly higher scores in the post-intervention on all social skills outcomes, and lower scores in the post-intervention on externalizing problems. In the case of the control group, significant differences were also observed between the two-time phases in social cooperation and social interaction, with higher scores in the post-intervention. However, the differences between the two phases were smaller than those observed in the experimental group. No significant differences were observed in social independence and externalizing problems in the control group. 

Figure 3 shows the pairwise contrast for the group*time interaction taking into account the means estimated by the model on social skills and externalizing problems variables.

Finally, the effect sizes between groups were calculated using the non-parametric Mann–Whitney test. The results showed a low effect size for externalizing problems (d = 0.24), medium effect size for social cooperation (d = 0.68) and social independence (d = 0.73), and high effect size for social interaction (d = 0.94).

## 4. Discussion

The main purpose of the research was to analyze the effectiveness of the AC1 program aimed at preschoolers aged three to four years. To this end, two objectives were established: (1) to test the effect of the program on the skills it trains; and (2) to analyze whether a transfer occurs and the program has an effect, in addition to the trained skills, on other social skills and externalizing problems.

Related to the first objective, the results showed that the participants in the experimental group achieved higher scores after the intervention in all the variables analyzed (rules, emotion IE, emotion regulation, communication skills, sharing and helping, sharing personal belongings, and solving problems). The control group also increased their scores in some variables.

The improvement in both groups in these skills is logical due to the socializing effect that school plays by facilitating social interactions and self-regulation, since children invest more time in relationships with peers [51], assume new rules of coexistence, and must adjust their behavior to the school context. However, even taking into account the initial differences between the groups, the gain was greater in the children who participated in the AC1 program. In addition, it should be noted that emotion regulation and sharing personal belongings only increased in the experimental group.

In this sense, the initial hypothesis that indicated that an improvement in the skills trained by the program was expected in the intervention group compared to the control group was confirmed. We can conclude that the AC1 program is effective and promotes the learning of social-emotional skills, with a greater effect on aspects related to the identification and expression of emotions, communication skills, solving problem, or prosocial behaviors such as sharing and helping. The improvement in these aspects is key for the person to develop an adjusted and self-regulated social-emotional behavior. Communication and conflict resolution skills are fundamental for maintaining healthy social interactions, and emotional understanding is essential for social adaptation and maintaining friendships [52].

The program has had a minor impact on the acquisition of rules (which may be logical due to the effect that the school itself exerts on this aspect) and on the regulation of emotions. Regarding the latter, we know that at three years old, children usually express the emotions they feel [53] and progressively learn to regulate, plan, supervise, and adjust their behavior to changing expectations and social circumstances, knowing that showing certain emotions may be appropriate in one situation, but not in another [54]. Although the effect of this variable has been weak, the children in the experimental group are the only ones who have made significant progress in controlling their emotions (mainly anger). We know that emotional regulation is a complex skill that requires a higher cognitive level, but the results indicate that its early development can be very positive, since without direct training, it is difficult to achieve notable progress at this age.

The results obtained were similar to those achieved by other preschool intervention programs such as the Socio-Emotional Program -SEP- [21], Al’s Pals: Kids Making Healthy Choices [55], or PATHS [32]. Specifically, Ştefan’s study, carried out with 158 4-year-old children, showed how the children in the experimental group (*n* = 89) improved their socio-emotional competencies compared to those in the control group, showing greater management in skills related to emotion identification, prosocial behaviors, and conflict resolution [21].

In relation to the second objective, we can see how the program had, not only, an effect on those aspects that it develops, but also a transfer and impact on other types of social skills and externalizing problems. The results of the pairwise contrast taking into account the estimated means showed an increase in social cooperation, social interaction, and social independence observed in the children of the experimental group as well as a reduction in the presence of externalizing problems. In contrast, the control group only increased their scores in cooperation and interaction, with no significant improvement in the rest of the variables. These results support the second hypothesis.

The program had a moderate effect on social cooperation and social independence and a high effect on social interaction. The development of prosocial behaviors such as sharing, helping or working in team as well as the assimilation of certain rules of harmony coexistence promoted by the AC1 program may have been key to improved social cooperation. On the other hand, the program also affects the development of communication skills, the search for solutions to problems, or the regulation of basic emotions, aspects that play an important role in the child’s autonomy. The effect of the program on cooperation and independence translates into greater regulation of behavior in social situations, as it has been observed that the children in the experimental group have improved in aspects such as the acceptance of rules, adaptation to group decisions or knowing how to control themselves, and feeling safe in different situations (they play with autonomously, play with different children, and accept separation from their parents).

The highest effect size was found in the social interaction variable. After participation in the program, the children in the experimental group showed a greater communicative capacity in class, they were more understanding with others, and were also warmer.

The results achieved in the social skills variables were consistent with those of other studies in which early intervention programs have been evaluated [32,56,57]. The Strong Start program was effective in improving the prosocial behaviors of 67 kindergarten students (5–6 years) after 35 sessions of 37 min duration [56].

The effectiveness of the Second Step program has also been analyzed in preschoolers. The study by Ocasio et al. [58] evaluated the impact of this program in 268 preschoolers aged 3–4 years old using the same instrument as in this research (PKBS-2). The results showed an improvement in cooperation, interaction, and social independence in all participants, although this time, it was not compared with a control group. The study by Kemple et al. [12] evaluated the effectiveness of the program using a control group, although with a smaller sample (*n* = 45) composed of children between three and four years of age. The results of the study showed an improvement in social knowledge, total social skills, cooperation, assertion, and self-control after 28 sessions in which contents related to empathy, emotion management, and problem-solving were worked on [12].

With respect to externalizing problems, the results showed that the experimental group significantly reduced their scores on this variable, as less aggressive behavior and minor attention problems were observed after the application of the AC1 program. Although the control group also showed a decreasing trend, the reduction was not the same as in the intervention group. Schooling has a positive effect and helps to reduce this type of problematic behavior. The rules that govern the school context, the directed work carried out at school, or the opportunities offered by this environment to establish relationships with peers, support the regulation of the children’s behavior. However, the results suggest that this acceleration in the reduction in externalizing problems in the experimental group may be due to the effect caused by the program.

In this sense, a small effect size was obtained in this variable, similar to those found in other intervention programs such as Projet Primar [59] or PATHS [32]. The effect size levels were lower for this variable. A similar finding and according to Ştefan [21], it is likely that changes in problem behaviors are more difficult to observe in the normalized population (used in this study), that is, in children who do not present a high-risk of behavioral problems.

The results obtained on the effectiveness of the program in reducing externalizing problems are in accordance with other studies already mentioned [12,32,55,58]. Specifically, SEP program significantly reduced externalizing problems in 89 children aged 4–5 years old after having participated in 37 sessions related to social competence (compliance with rules, prosocial behavior, and problem solving) and emotional competence (recognition and regulation) [60].

Finally, we must highlight a series of limitations of the research, which should be taken into account when interpreting the magnitude of the results obtained. The first refers to the origin of the participant sample, which makes external validity difficult. However, we can highlight other studies that have evaluated the efficacy of the AC1 program in other schools in Granada [40,41,42,61,62] or in other cities of Spain [63]. The program is also being adapted and validated in different Latin American countries thanks to research projects.

Another limitation would be associated with the evaluation, which was reduced in this case to a single external evaluator. It would be interesting to carry out a hetero-evaluation and include observations provided by parents and teachers, as has been done in other studies [21].

Finally, it would also be important to analyze the long-term effects of the program. Some longitudinal studies have been carried out over three years of intervention [42], but it would be desirable to include a follow-up phase to analyze this type of effect afterward including a cost–benefit analysis of the intervention.

## 5. Conclusions

In consonance with the results found, we conclude that the AC1 program could be beneficial in promoting the learning of social-emotional skills that facilitate regulated behavior, reducing the presence of externalizing problems. However, we consider that it would be important to include specific strategies focused on the reduction in behavioral problems that more directly involve behavioral regulation in prevention programs. Social-emotional competence and self-regulation are two closely related but distinct constructs [64] and children with behavioral problems often present difficulties with the management of self-regulation skills [7].

Despite the limitations noted above, the research presents a number of strengths and implications for professionals working in schools. The program can be of great utility to teachers as it is easily applicable and can also be inserted into the school curriculum. Since it is a universal intervention, it can be implemented at other schools and reach more preschoolers, thus acquiring a preventive character.

The results obtained, together with those achieved by other intervention programs, contribute to the growth of a research body that supports the inclusion of social-emotional learning in the preschool curriculum. Social-emotional competence in early childhood is fundamental for the positive development of the person [65], and as we have argued, this type of competence is closely related to regulated adaptive behavior and helps to reduce the presence of behavioral problems.

## Figures and Tables

**Figure 1 ijerph-18-08447-f001:**
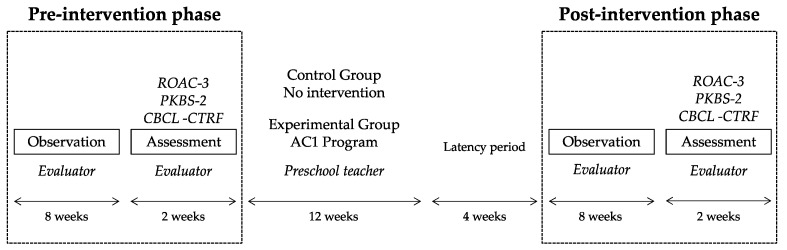
Study phases.

**Figure 2 ijerph-18-08447-f002:**
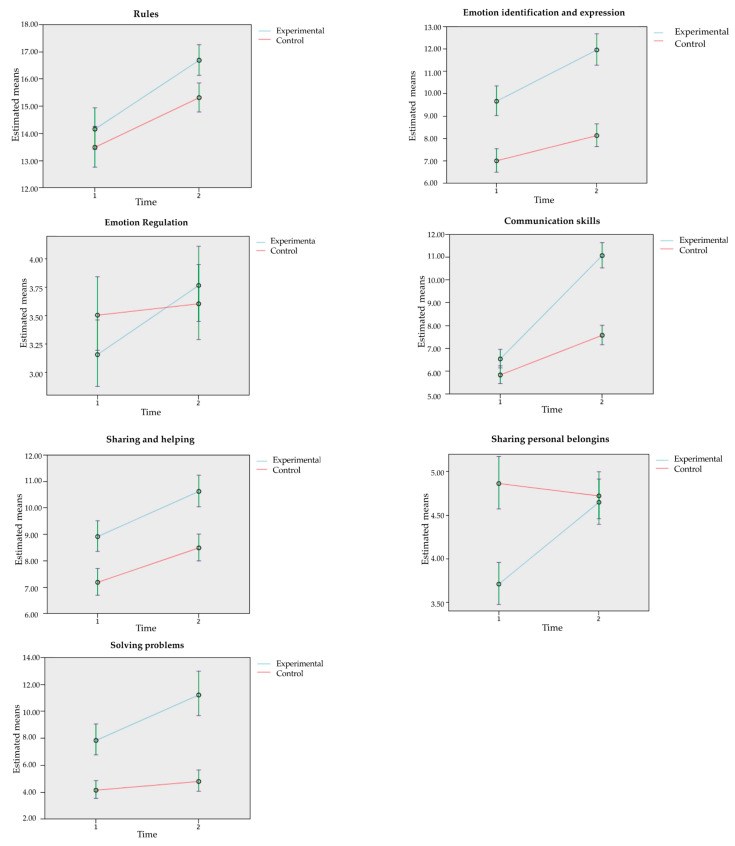
Pairwise contrast in the group*time interaction using the model-estimating means for the program’s variables.

**Figure 3 ijerph-18-08447-f003:**
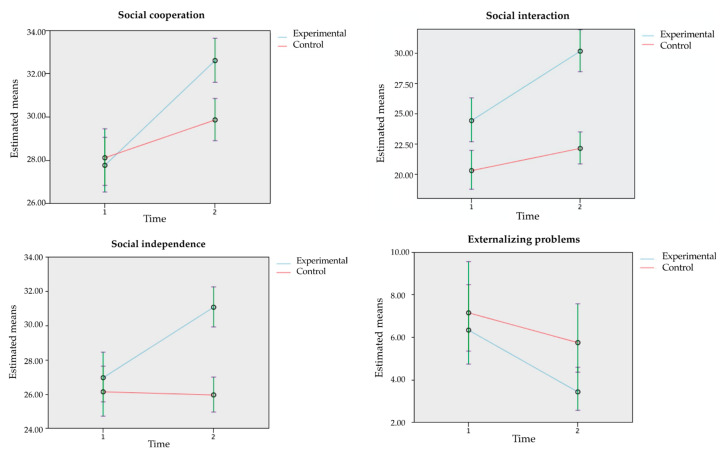
Pairwise contrast in the group*time interaction using the model-estimating means for the program’s variables.

**Table 1 ijerph-18-08447-t001:** Descriptive measures related to the variables used in AC1 program by groups and phases.

Outcome Variable	Pre-Intervention			Post-Intervention		
	M (SD)	Min.	Max.	M (SD)	Min.	Max.
G. Experimental						
Rules	14.25 (0.36)	8.00	18.00	16.38 (0.17)	14.00	18.00
Emotion IE	9.65 (0.21)	6.00	12.00	11.96 (0.27)	11.00	12.00
Emotion regulation	3.18 (0.14)	1.00	4.00	3.82 (0.13)	2.00	6.00
Communication skills	6.55 (0.18)	4.00	9.00	11.08 (0.15)	8.00	12.00
Sharing and helping	8.92 (0.18)	6.00	12.00	10.62 (0.17)	8.00	12.00
Sharing personal belongings	3.70 (0.13)	2.00	5.00	4.67 (0.12)	2.00	6.00
Solving problems	7.88 (0.23)	5.00	12.00	11.29 (0.16)	8.00	12.00
G. Control						
Rules	13.61 (0.39)	8.00	18.00	15.49 (0.33)	10.00	18.00
Emotion IE	7.24 (0.36)	2.00	12.00	8.41 (0.37)	5.00	12.00
Emotion regulation	3.57 (0.17)	1.00	6.00	3.67 (0.20)	1.00	6.00
Communication skills	5.90 (0.21)	2.00	8.00	7.65 (0.28)	4.00	11.00
Sharing and helping	7.33 (0.34)	2.00	12.00	8.65 (0.33)	3.00	12.00
Sharing personal belongings	4.90 (0.15)	3.00	6.00	4.75 (0.14)	3.00	6.00
Solving problems	4.96 (0.42)	0.00	9.00	5.73 (0.55)	0.00	12.00

Note. M = Mean, SD = Standard deviation.

**Table 2 ijerph-18-08447-t002:** Descriptive dates in social skills and externalizing problems by groups and phases.

Outcome Variable	Pre-Intervention			Post-Intervention		
	M (SD)	Min.	Max.	M (SD)	Min.	Max.
G. Experimental						
Social cooperation	27.83 (0.72)	11.00	36.00	32.67 (0.35)	28.00	36.00
Social interaction	24.54 (0.80)	9.00	33.00	30.29 (0.36)	23.00	33.00
Social independence	27.04 (0.79)	13.00	33.00	31.15 (0.34)	25.00	33.00
Externalizing problems	7.90 (1.10)	0.00	35.00	4.28 (0.55)	0.00	16.00
G. Control						
Social cooperation	28.24 (0.56)	19.00	34.00	30.00 (0.60)	19.00	36.00
Social interaction	21.04 (0.97)	6.00	31.00	22.94 (0.89)	10.00	33.00
Social independence	26.41 (0.69)	16.00	33.00	26.24 (0.64)	15.00	33.00
Externalizing problems	7.39 (1.24)	0.00	33.00	7.55 (1.12)	0.00	27.00

Note. M = Mean, SD = Standard deviation.

**Table 3 ijerph-18-08447-t003:** Results of generalized linear mixed models estimating change in the program’s outcomes.

Outcome Variable	*F*	Coefficient (SE)	*t*	*p*-Value	CI 95%	
					Inferior	Superior
Rules						
Group	5.66 *	0.09 (0.02)	3.50	0.001	0.04	0.13
Time	77.39 ***	−0.13 (0.02)	−5.34	<0.001	−0.17	−0.08
Group*Time	1.29	−0.04 (0.03)	−1.14	0.257	−0.10	0.02
Emotion IE						
Group	65.18 ***	0.38 (0.04)	8.87	<0.001	0.30	0.47
Time	95.32 ***	−0.15 (0.03)	−5.24	<0.001	−0.20	−0.09
Group*Time	2.94	−0.06 (0.04)	−1.72	0.088	−0.14	0.01
Emotion regulation						
Group	0.29	0.04 (0.06)	0.68	0.496	−0.08	0.17
Time	9.17 **	−0.03 (0.05)	−0.59	0.558	−0.12	0.07
Group*Time	4.81 *	−0.15 (0.07)	−2.19	0.029	−0.28	−0.01
Communication skills						
Group	47.19 ***	0.38 (0.04)	9.97	<0.001	0.30	0.45
Time	317.30 ***	−0.26 (0.03)	−7.89	<0.001	−0.33	−0.19
Group*Time	36.22 ***	−0.27 (0.04)	−6.02	<0.001	−0.35	−0.18
Sharing and helping						
Group	29.85 ***	0.22 (0.04)	5.39	<0.001	0.14	0.31
Time	68.15 ***	−0.17 (0.03)	−5.35	<0.001	−0.23	−0.10
Group*Time	0.04	−0.01 (0.04)	−0.21	0.833	−0.09	0.07
Sharing personal belongings						
Group	15.28 ***	−0.02 (0.04)	−0.39	0.699	−0.09	0.06
Time	18.75 ***	0.03 (0.03)	0.95	0.341	−0.03	0.09
Group*Time	31.82 ***	−0.25 (0.04)	−5.64	<0.001	−0.34	−0.17
Solving problems						
Group	49.44 ***	0.85 (0.11)	7.59	<0.001	0.63	1.07
Time	61.33 ***	−0.15 (0.05)	−2.79	0.006	−0.25	−0.04
Group*Time	10.96 **	−0.21 (0.06)	−3.11	0.001	−0.34	−0.09

Note. DF = 198, SE = Standard error, CI = Confidence intervals, * *p* < 0.01, ** *p* < 0.05, *** *p* < 0.001.

**Table 4 ijerph-18-08447-t004:** Pairwise contrast of groups and times using means estimated by the model on the program’s variables.

Outcome Variable	Estimation (SD)	*t*	*p*−Value	CI 95%	
				Inferior	Superior
Rules					
Group (Exp-Cont.)	0.98 (0.42)	2.38	0.018	0.17	1.82
Time (pre-pos)	−2.17 (0.23)	−9.30	<0.001	−2.62	−1.71
Emotion IE					
Group (Exp-Cont.)	3.20 (0.40)	7.95	< 0.001	2.41	3.99
Time (pre-pos)	−1.62 (0.16)	−10.01	< 0.001	−1.95	−1.31
Emotion regulation					
Group (Exp-Cont.)	−0.11 (0.19)	−0.54	0.588	−0.49	0.628
Time (pre-pos)	−0.36 (0.12)	−3.02	< 0.003	−0.59	−0.12
Communication skills					
Group (Exp-Cont.)	1.86 (0.27)	6.84	<0.001	1.32	2.39
Time (pre-pos)	−2.98 (0.35)	−17.86	<0.001	−3.31	−2.65
Sharing and helping					
Group (Exp-Cont.)	1.92 (0.35)	5.44	<0.001	1.22	2.62
Time (pre-pos)	−1.49 (0.18)	−8.38	<0.001	−1.84	−1.14
Sharing personal belongings					
Group (Exp-Cont.)	−0.64(0.16)	−3.89	<0.001	−0.96	−0.31
Time (pre-pos)	−0.44 (0.10)	−4.36	<0.001	−0.63	−0.24
Solving problems					
Group (Exp-Cont.)	4.90 (0.75)	6.50	<0.001	3.42	6.39
Time (pre-pos)	−1.63 (0.23)	−7.09	<0.001	−2.09	−1.18

Note. SD = Standard deviation, CI = Confidence intervals.

**Table 5 ijerph-18-08447-t005:** Pairwise contrast of the group*time interaction, using means estimated by the model on the program’s variables.

Outcome Variable	Estimation (SD)	*t*	*p*−Value	CI 95%	
				Inferior	Superior
Rules					
Experimental (pre-pos)	−2.53 (0.33)	−7.58	<0.001	−3.19	−1.87
Control (pre-pos)	−1.82 (0.32)	−5.61	<0.001	−2.46	−1.18
Emotion IE					
Experimental (pre-pos)	−2.29 (0.25)	−9.14	<0.001	−2.78	−1.79
Control (pre-pos)	−1.12 (0.21)	−5.37	<0.001	−1.54	−0.71
Emotion regulation					
Experimental (pre-pos)	−0.61 (0.16)	−3.69	< 0.001	−0.93	−0.28
Control (pre-pos)	−0.10 (0.17)	−0.59	0.558	−0.44	0.24
Communication skills					
Experimental (pre-pos)	−4.53 (0.25)	−17.75	<0.001	−5.03	−4.03
Control (pre-pos)	−1.74 (0.22)	−7.97	<0.001	−2.17	−1.31
Sharing and helping					
Experimental (pre-pos)	−1.71 (0.26)	−6.51	<0.001	−2.22	−1.19
Control (pre-pos)	−1.30 (0.24)	−5.44	<0.001	−1.77	−0.83
Sharing personal belongings					
Experimental (pre-pos)	−0.94 (0.13)	−6.93	<0.001	−1.20	−0.67
Control (pre-pos)	0.14 (0.15)	0.95	0.343	−0.15	0.44
Solving problems					
Experimental (pre-pos)	−3.37 (0.44)	−7.72	<0.001	−4.24	−2.51
Control (pre-pos)	−0.65 (0.24)	−2.69	0.008	−1.13	−0.17

Note. SD = Standard deviation, CI = Confidence intervals.

**Table 6 ijerph-18-08447-t006:** Results of generalized linear mixed models estimating change in outcomes social skills and externalizing problems.

Outcome Variable	*F*	Coefficient (SE)	*t*	*p*-Value	CI 95%	
					Inferior	Superior
Social cooperation						
Group	2.44	3.40 (0.02)	3.84	<0.001	0.04	0.13
Time	54.55 ***	−0.06 (0.02)	−2.81	0.005	−0.10	−0.02
Group*Time	11.25 **	−0.10 (0.03)	−3.35	0.001	−0.16	−0.04
Social interaction						
Group	30.03 ***	0.31 (0.04)	7.36	<0.001	0.23	0.39
Time	62.38 ***	−0.09 (0.03)	−3.08	0.002	−0.14	−0.03
Group*Time	10.91 **	−0.12 (0.04)	−3.30	0.001	−0.20	0.05
Social independence						
Group	12.49 **	0.18 (0.03)	6.53	<0.001	0.12	0.23
Time	17.57 ***	0.01 (0.02)	0.30	0.765	−0.04	0.05
Group*Time	21.40 ***	−0.15 (0.03)	−4.63	<0.001	−0.21	−0.08
Externalizing problems						
Group	2.81	−0.51 (0.20)	−2.54	0.012	−0.91	−0.11
Time	27.12 ***	0.22 (0.10)	2.09	0.038	0.01	0.42
Group*Time	6.11 *	0.39 (0.16)	2.47	0.014	0.08	0.71

Note. DF = 198, SE = Standard error, CI = Confidence intervals, *p* < 0.01, ** *p* < 0.05, *** *p* < 0.001.

**Table 7 ijerph-18-08447-t007:** Pairwise contrast of groups and times using means estimated by the model on social skills and externalizing problems.

Outcome Variable	Estimation (SD)	*t*	*p*-Value	CI 95%	
				Inferior	Superior
Social cooperation					
Group (Exp-Cont.)	1.11 (0.71)	1.56	0.120	−0.92	2.51
Time (pre-pos)	−3.26 (0.41)	−7.62	<0.001	−4.11	−2.42
Social interaction					
Group (Exp-Cont.)	5.94 (1.09)	5.44	<0.001	3.79	8.10
Time (pre-pos)	−3.57 (0.43)	−8.31	<0.001	−4.41	−2.72
Social independence					
Group (Exp-Cont.)	2.90 (0.82)	3.53	<0.001	1,28	4.51
Time (pre-pos)	−1.85 (0.43)	−4.29	<0.001	−2.70	−1.00
Externalizing problems					
Group (Exp-Cont.)	−1.74 (1.06)	−1.64	0.102	−3.84	0.35
Time (pre-pos)	2.29 (0.50)	4.56	<0.001	1.30	3.28

Note. SD = Standard deviation, CI = Confidence intervals.

**Table 8 ijerph-18-08447-t008:** Pairwise contrast of the group*time interaction, using means estimated by the model on social skills and externalizing problems.

Outcome Variable	Estimation (SD)	*t*	*p*-Value	CI 95%	
				Inferior	Superior
Social cooperation					
Experimental (pre-pos)	−4.84 (0.60)	−8.06	<0.001	−6.02	−3.65
Control (pre-pos)	−1,75 (0.61)	−2.86	0.005	−2.95	−0.54
Social interaction					
Experimental (pre-pos)	−5.73 (0.64)	−8.91	<0.001	−6.99	−4.46
Control (pre-pos)	−1.83 (0.57)	−3.19	<0.002	−2.97	−0.70
Social independence					
Experimental (pre-pos)	−4.10 (0.61)	−6.70	<0.001	−5.31	−2.90
Control (pre-pos)	0.18 (0.61)	0.30	0.765	−1.02	1.38
Externalizing problems					
Experimental (pre-pos)	2.90 (0.71)	4.11	<0.001	1.51	4.29
Control (pre-pos)	1.40 (0.71)	1.97	0.051	−0.01	2.80

Note. SD = Standard deviation, CI = Confidence intervals.

## Data Availability

Data are available on request due to ethical restrictions.

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
