# Peer review of "Can We Manage Behavioral Problems through the Development of Children’s Social-Emotional Regulated Behavior? Longitudinal Study of a Preschool Program"

_ijerph, 2021, doi:10.3390/ijerph18168447_

Round 1

Reviewer 1 Report

I am very grateful to the editor for the invitation to review the article “Can we Manage Behavioral Problems through the Development of Children’s Social-Emotional Regulated Behavior? Longitudinal Study of a Preschool Program ”. As the authors indicate, there is little evidence on the object of study in Spain, and I consider this work a  good contribution to the line of research. 

The structure of the article is correct, and the methodology, perfectly explained, yields highly significant results. The discussion section is well planned and specifies what this research contributes to the field of knowledge in comparison with previous studies. Moreover, the limitations also shed light on the study. Finally, the conclusions are clearly justified from the results of the work. 

Although I consider that the article can be perfectly published in its current version, I wonder if the data provided in the lines 43-48 could be, as far as possible, more recent. Likewise, I would like to pose a question: as the authors indicate, behavioral problems may respond, among others, to contextual factors, eg. poverty, does it have any effect that in the sample all the participants belong to the same socioeconomic context? 

Reviewer 2 Report

Anna Justicia Arraez et al have conducted a very interesting study on evaluating the effect of AC1  preschool program on social-emotional skills of children. It is a very interesting study well conducted. I have some minor comment : 

  1. The Introduction section is huge for a original paper. It needs to be summarized to the essential aspects of the problem on study. Moreover, AC1 program which was applied in the study was not mentioned in the Introduction. A brief report on the how efficient was the application of this program in other countries needs to be added. 
  2. Who was responsible for the training of the teacher of the AC1 program? This needs to be added in the Methodology section. Did the teacher pass an exam which certifies this education? 
  3. In tables 1,2 the p values of differences between pre and post-intervention program and between experimental and controls using the appropriate statistical tests need to be added. 
  4. I don’t understand the utility of generalized linear mixed model in this study since only two longitudinal measurements for each variable were taken. Why not compare the variables between the two groups (exper/cont) at two time points (before and after intervention) and in each group separately before and after intevention (paired test)? Which more information was given by the generalized linear mixed model? 

Round 2

Reviewer 2 Report

No further comments